# Leveraging Federated Satellite Systems for Unmanned Medical Evacuation on the Battlefield

**DOI:** 10.3390/s25061655

**Published:** 2025-03-07

**Authors:** Kasper Halme, Oskari Kirjamäki, Samuli Pietarinen, Mikko Majanen, Kai Virtanen, Marko Höyhtyä

**Affiliations:** 1Department of Military Technology, Finnish National Defence University, 00861 Helsinki, Finland; 2Safe and Connected Society, VTT Technical Research Centre of Finland Ltd., 02044 Espoo, Finland; 3Department of Mathematics and Systems Analysis, Aalto University, 00076 Espoo, Finland

**Keywords:** federated systems, medical evacuation, satellites, simulation, unmanned vehicles

## Abstract

This paper evaluates the role of federated satellite systems (FSSs) in enhancing unmanned vehicle-supported military medical evacuation (MEDEVAC) missions. An FSS integrates multiple satellite systems, thus improving imaging and communication capabilities compared with standalone satellite systems. A simulation model is developed for a MEDEVAC mission where the FSS control of an unmanned aerial vehicle is distributed across different countries. The model is utilized in a simulation experiment in which the capabilities of the federated and standalone systems in MEDEVAC are compared. The performance of these systems is evaluated by using the most meaningful metrics, i.e., mission duration and data latency, for evacuation to enable life-saving procedures. The simulation results indicate that the FSS, using low-Earth-orbit constellations, outperforms standalone satellite systems. The use of the FSS leads to faster response times for urgent evacuations and low latency for the real-time control of unmanned vehicles, enabling advanced remote medical procedures. These findings suggest that investing in hybrid satellite architectures and fostering international collaboration promote scalability, interoperability, and frequent-imaging opportunities. Such features of satellite systems are vital to enhancing unmanned vehicle-supported MEDEVAC missions in combat zones.

## 1. Introduction

In modern military operations, where seconds mean the difference between life and death, the ability to efficiently conduct medical evacuation (MEDEVAC) missions under hostile conditions is vital. MEDEVAC missions are classified into three levels of urgency: urgent evacuations for life-threatening injuries that require intervention within one hour, priority evacuations for severe but non-life-threatening conditions with a response window of up to four hours, and routine evacuations permitting delays of up to twenty-four hours [1]. Traditional MEDEVAC practices, such as manned ambulances, helicopters, or boats, remain essential to evacuating casualties from conflict zones but encounter significant operational challenges. Hazardous terrain, including mountains, forests, and urban warfare zones, can obstruct or delay evacuation efforts. These difficulties are even more pronounced in remote or poorly mapped areas [2]. Additionally, MEDEVAC personnel are highly vulnerable, as manned vehicles are prime targets for adversaries, placing rescuers and casualties at risk [3,4]. Resource constraints further compound these challenges, as the high operational costs of sustaining MEDEVAC teams and vehicles, combined with limited availability in prolonged or large-scale conflicts, significantly reduce overall evacuation capacity [4].

Unmanned vehicle systems (UXVs), particularly aerial platforms such as drones and unmanned helicopters, are proving transformative in overcoming the operational challenges of traditional MEDEVAC means [5,6,7]. One of the key advantages of UXVs is their ability to navigate terrain impassable for ground vehicles or helicopters. Aerial UXVs can bypass obstacles like mountains or urban areas to reach injured personnel faster [8,9]. In addition to improving accessibility, UXVs minimize risks to human personnel by taking on dangerous tasks, like navigating minefields, operating in areas exposed to enemy fire, or delivering supplies through hazardous terrain. Instead of deploying a human crew into a conflict zone for casualty evacuation, a UXV can perform the mission, reducing exposure to threats like ambushes or improvised explosive devices [3,4]. Additionally, UXVs are cost-effective in the long term, as they decrease reliance on manned vehicles and the associated costs of maintaining human personnel and infrastructure [9].

Satellite systems enhance UXV capabilities by providing key functionalities based on their payloads. For instance, imaging satellites provide terrain analysis and, together with global navigation satellite systems (GNSSs), enable efficient route planning, threat avoidance, and the identification of suitable landing zones for UXVs. These enhancements reduce mission duration and fuel consumption, thereby minimizing operational delays [8,9,10]. Communication satellites facilitate seamless data transfer between a medical command (MEDCOM) and UXVs. This functionality supports the real-time transmission of video feeds and sensor data throughout mission phases. This capability allows MEDCOM to assess casualties and prioritize evacuations to appropriate medical facilities [7].

While satellite integration enhances UXV-supported MEDEVAC missions, standalone satellite systems (SSSs) face the challenges of having small networks and restricted satellite-to-satellite communication capabilities. Since imaging satellite data must pass through ground stations before reaching UXVs, transmission delays can disrupt real-time navigation, limiting the satellites’ ability to track locations and adjust paths based on new information. For instance, it may take hours or even days to download and analyze satellite data at a MEDCOM. Delays from satellite alignment with ground stations can impact the control decisions of UXVs, such as rerouting or adjusting positions. These delays are frequent in remote or hostile areas with limited ground station coverage, making satellites impractical in such circumstances [11]. Additionally, inefficiencies like high data latency from distant satellites, limited bandwidth during large-scale evacuations, and restricted imaging capabilities (e.g., single-angle views) prolong MEDEVAC missions and directly affect evacuee survival rates [12,13,14,15]. Finally, each SSS requires distinct infrastructure for control and data access, complicating the use of multiple satellite systems during MEDEVAC missions [16,17,18].

To address these challenges, federated satellite systems (FSSs) present a promising alternative to SSSs capable of utilizing multiple satellite systems [19]. This paper defines an FSS as a collaborative framework in which numerous imaging satellites from different organizations and nations are combined, and any data from any satellite can be accessed from a standard user interface. The FSS may include inter-satellite links (ISLs) for faster data relay, and data processing capabilities are shared among the accessing organizations. Like NATO’s advanced persistent surveillance system (APSS), the FSS encourages inter-satellite cooperation. This collaboration enables rapid adaptation to shifting conditions and mission needs while ensuring satellites are used effectively [20]. For instance, in a military operation, the FSS enables satellites to adjust their coverage areas—the regions they can observe or communicate with—based on real-time intelligence. If a UXV encounters an obstacle or a casualty needing evacuation, the satellite system can adjust the satellite’s sensors, such as a camera, to assist with the UXV’s navigation. However, unlike the APSS, which is designed for NATO’s mission-specific needs and relies on predefined partnerships, the FSS prioritizes broader interoperability and resource sharing across diverse missions and stakeholders. Both systems reduce infrastructure needs, but the FSS is more flexible, allowing for on-demand satellite collaborations.

Table 1 summarizes the benefits of the FSS compared with SSSs. An FSS offers two significant improvements over an SSS. First, the increased number of satellites within the system allows for more frequent and detailed observations of the desired target. Second, transmitting data via ISLs facilitates faster transmission to ground stations. In the FSS, satellites owned by different partners collaborate through standard user interfaces, providing authenticated users with integrated and real-time data access. The FSS mainly combines various imaging systems and then utilizes communication and positioning systems alongside them. Ground stations manage data processing and maintain secure communication, crucial to high-risk missions like MEDEVAC. By using multiple satellite perspectives, the FSS enhances situational awareness with real-time data and improves the deployment of UXVs through precise navigation and communication, thereby reducing delays during casualty evacuations [17].

Existing studies on FSSs focus on supporting applications, for example, automotive, energy, and commercial UXV operations [21], often overlooking aspects like real-time data integration, quick decision making, and seamless interoperability—key requirements for effective MEDEVAC missions [8,9,22]. Although some studies have explored the idea of combining SSSs with MEDEVAC, the integration of an FSS within the UXV-supported MEDEVAC context remains unexamined [7,8,9,10]. Furthermore, it is crucial to understand how SSSs perform in this context. This paper addresses these gaps in the existing literature by conducting a simulation experiment to evaluate and compare the performance of an FSS and SSSs during a MEDEVAC mission.

The MEDEVAC mission considered in this paper is defined as a two-phase operation. In the first phase, the MEDCOM receives information about a casualty and employs imaging satellites to capture and transmit location data via ISLs. In the second phase, a UXV is dispatched and navigated by using satellite-assisted navigation while the MEDCOM monitors its movement in real time. The mission concludes when the UXV reaches the casualty. In practice, the mission would proceed with triage, followed by a handover to medical personnel for advanced care.

In this paper, a simulation model for the MEDEVAC mission is developed. The model considers the main elements of the mission and the interactions between the FSS, the SSSs, and the UXV. Generally, simulations offer a cost-effective and flexible approach to analyzing systems under various conditions, such as hostile interference [23,24,25,26]. There are several tools for the simulation of satellite systems, including NASA’s general mission analysis tool, which specializes in orbital dynamics and mission analysis [27]. However, the simulation model for the MEDEVAC mission is constructed by using Systems Tool Kit 12.10 (STK) simulation software, which provides methods for analyzing the performance of satellites [28].

In the simulation experiment, five MEDEVAC missions are analyzed, each utilizing a specific satellite system: one using an FSS and four using an SSS. The evaluation of these satellite systems depends on two key metrics: The first metric is mission duration, which measures the time from identifying the need for MEDEVAC to the UXV’s arrival at the casualty. The second metric is data latency, which quantifies communication delays between the MEDCOM and the UXV. The values of these metrics obtained in the simulations classify the alternative satellite systems according to the types of MEDEVAC missions they can support. The findings from the simulation experiment reveal the strengths and weaknesses of these systems and ultimately emphasize the superiority of the FSS in supporting UXVs during MEDEVAC missions.

The paper is structured as follows: Section 2 outlines the MEDEVAC mission, detailing its phases and elements. Section 3 describes the simulation model and performance metrics. Section 4 presents the results of the simulation. Section 5 discusses the findings and proposes directions for future research, while Section 6 presents the conclusions.

## 2. The MEDEVAC Mission

### 2.1. Phases of the MEDEVAC Mission

The MEDEVAC mission considered in the simulation experiment consists of two phases, as illustrated in Figure 1. The first phase, presented in Figure 1a, begins with the MEDCOM receiving initial information about a casualty and requesting that the imaging satellite adjust its camera to capture an image of the casualty’s location. As the satellite flies over the region, it captures the image. The data are then transmitted to the MEDCOM via a ground station, utilizing ISLs whenever available to expedite data transfer. This phase establishes the geospatial foundation for the mission by confirming the casualty’s location.

In the second phase, illustrated in Figure 1b, the MEDCOM dispatches a UXV from the launch site to the casualty after receiving the data. The UXV is connected to the MEDCOM via communication satellites. The MEDCOM monitors the UXV’s position, speed, and operational condition in real time, ensuring that the mission progresses smoothly. The MEDEVAC mission concludes in the simulation experiment once the UXV arrives at the casualty’s location.

### 2.2. Elements of the MEDEVAC Mission

The MEDEVAC mission consists of seven key elements: the environment, a casualty, the MEDCOM, ground stations, the launch site, a UXV, and the satellite system. The mission environment is set for 1 November 2024, from 08:00 to 20:00, with satellites positioned and moving precisely as they would. The weather conditions are suitable for capturing the image, with no clouds present, ensuring that the photo can be successfully taken on the first attempt. The casualty, located near the Finnish–Russian border, is a critically injured individual in urgent need of medical assistance.

The MEDCOM located in Brussels, Belgium, is the central hub for coordinating the MEDEVAC mission. With advanced communication systems, the MEDCOM facilitates seamless communication between UXVs and satellite systems. It utilizes satellite imagery to monitor the mission’s progress, track casualties, and adjust plans as necessary, ensuring timely and effective evacuations.

The satellite ground stations act as nodes for receiving, processing, and relaying satellite data to the MEDCOM. The MEDEVAC mission includes five imaging satellite ground stations: Kiruna, Toulouse, Weilheim, Neustrelitz, and Cordoba. The Kiruna station, in Sweden, recognized for its strategic location above the Arctic Circle, facilitates frequent passes of polar-orbiting satellites [29]. The Toulouse station, in France, a hub for European space activities, plays a crucial role in satellite operations and data analysis [30]. In Germany, the Weilheim and Neustrelitz stations provide robust infrastructure for satellite telemetry, with Neustrelitz specializing in Earth observation data [31]. The Cordoba station, in Argentina, enhances global imaging coverage, aiding in data reception and dissemination in the Southern Hemisphere [32].

The launch site, located near the eastern border of Finland at Lappeenranta Airport, provides a strategically advantageous position for launching the UXV. The airport’s well-established infrastructure, which includes a 2500 m long runway and extensive airside facilities, facilitates the seamless integration of unmanned ground and aerial systems. Furthermore, its controlled airspace and advanced communication systems support real-time coordination between the UXV and the MEDCOM.

The UXV is designed for long-range MEDEVAC missions, featuring advanced capabilities inspired by platforms such as the Grille UAV, the K-MAX helicopter, and the BAE Systems’ T-650 heavy-lift drone [33,34,35]. With a maximum speed of 190 km/h and a 500–700 kg payload capacity, the UXV offers telesurgery functionalities, including real-time video capabilities.

Five different satellite systems are used, labeled ‘France’, ‘Germany’, ‘Italy’, ‘EU’, and ‘Federated’. As Table 2 illustrates, each satellite system comprises imaging satellites, communication satellites, and ground stations. The ‘France’, ‘Germany’, ‘Italy’, and ‘EU’ systems function as SSSs, each utilizing dedicated satellites for imaging and data transfer. In contrast, the ‘Federated’ system connects satellites to enhance data sharing and coordination among all satellite systems. The ‘France’ system employs the Pleiades [36] constellation for imaging and the Syracuse 3A and 3B satellites for secure data transfer [37]. The ‘Germany’ system uses the PAZ, TerraSAR-X, and TanDEM-X satellites for imaging [38,39] and the COMSATBw-1 and COMSATBw-2 satellites for data transfer [40]. The ‘Italy’ system utilizes the COSMO-SkyMed constellation for imaging [41] and the Athena-Fidus satellite for data transfer [42]. The ‘EU’ system uses the Sentinel-2 constellation for imaging [43], along with the EUTELSAT OneWeb constellation for data transfer [44]. The ‘Federated’ system integrates ISL-enabled satellites for imaging, allowing for multiple data relay routes. This system explores the potential benefits of inter-satellite communication, though many real-world imaging satellites presently lack such capabilities. All imaging satellites are in low Earth orbit (LEO), while all communication satellites are in geostationary orbit (GEO), except for OneWeb, which is in LEO.

## 3. Simulation of MEDEVAC Mission

### 3.1. The Simulation Model

The simulation model of the MEDEVAC mission is constructed by using STK software [28], which allows for the precise modeling of satellite systems and their interactions, including satellites, ground stations, and communication links. STK enables users to simulate satellite orbits, signal coverage, and sensor performance while generating performance reports on metrics like data latency. In STK, objects represent real-world entities, such as satellites, ground stations, or aircraft, each with specific attributes, namely, orbital paths or coverage areas. These objects interact and are often arranged hierarchically, enabling detailed analysis of satellite line of sight, signal coverage, and collision risks. The MEDEVAC model employs standard STK objects to represent the mission accurately. The model is deterministic, yielding the same outcomes for identical inputs. However, since satellite positions depend on the starting time of a simulation run, variations in initial conditions can lead to differences in simulation results.

The elements of the MEDEVAC mission discussed in Section 2.2 are described by using STK objects. The mission environment is defined as a ‘Scenario’ object, including time, weather, and daylight parameters. The MEDCOM is presented as a ‘Place’ object with geospatial coordinates. The casualty’s location is represented as a ‘Target’ object, serving as the focal point for satellite imaging and the navigation of the UXV. The UXV is simulated as an ‘Airplane’ object, which allows for the representation of its flight path, including its position and speed. The UXV flies directly from the launch site to the casualty at 190 km/h. The launch site for the UXV is identified as a ‘Place’ object. Satellites and ground stations are imported from the Ansys Government Initiatives (AGI) database via STK, allowing for the retrieval of various parameters, for example, orbital positions, signal range, communication capabilities, and other satellite-specific characteristics used in the simulation [28]. Satellites are primarily modeled as ‘Satellite’ objects configured with real-world orbits and communication capabilities. At the same time, the OneWeb constellation is represented as a ‘Satellite Collection’ to handle its network of satellites. Ground stations are depicted as ‘Facility’ objects with locations and communication parameters.

To ensure consistent communication, only the orbits and positions of the selected satellite systems and ground stations are modeled. The communication link is always assumed to be available when there is a direct line-of-sight connection between the ground station and the satellite, the MEDCOM and the satellite, or the UXV and the satellite. Additionally, for communication between ground stations and the satellites, the elevation angle has to be over 10° above the horizon for the imaging satellites and over 30° for the communication satellites.

As mentioned above, the attributes of the satellite object are sourced from the AGI database, providing high-quality and validated inputs for the simulation model. For the Sentinel satellites, the half-angle of their rectangular sensor is set to 21°, and the horizontal half-angle is configured to 3.5°, since these values are not available in the AGI database [45]. Other satellites, such as CSG-1 and CSG-2, part of the ‘Italy’ system, are adjusted to reflect COSMO-SkyMed1 specifications, ensuring consistency with real-world performance characteristics. Similarly, the PAZ satellite in the ‘Germany’ system is configured by using attributes derived from TerraSAR-X to address gaps in the AGI database and maintain a valid representation of its capabilities.

Several steps are taken to verify and validate the simulation model. The configurations of the satellite objects are cross-checked against publicly available specifications and mission documentation. The performance of satellite sensors, including field-of-view and resolution, is evaluated through line-of-sight and coverage analysis tools in STK.

### 3.2. The Performance Metrics

Performance metrics are used to evaluate the operational efficiency of the SSSs and the FSS in supporting the use of a UXV for a MEDEVAC mission. The metrics used in the simulation experiment are the duration of the mission and the data transmission latency, which are introduced next. Different types of MEDEVAC missions performed with UXVs require specific mission duration and latency values. The following sections present how the missions are classified based on these metrics.

#### 3.2.1. The Mission Duration Metric

The mission duration metric represents the total time required to complete the MEDEVAC mission and consists of two parts: The first part measures the time from the MEDCOM’s request for an image of the casualty to the reception of that image during the first phase of the mission (see Figure 1a). This duration encompasses the communication between the satellite and the ground station and the revisit time of the satellite (i.e., the frequency with which it passes over the casualty). The second part of the mission duration metric addresses the UXV’s travel time to the casualty’s location in the second phase of the mission. This location is approximately 117 km from the launch site. With a constant speed of 190 km/h, as outlined in Section 2.2, the travel time is calculated to be 37 min and 4 s.

The urgency of missions is classified into the ‘Urgent’, ‘Priority’, or ‘Routine’ class based on the duration, as detailed in Table 3 [1]. ‘Urgent’ missions, which must be completed within one hour, aim to prevent serious complications or permanent damage to casualties and require the quickest response. ‘Priority’ missions, needing completion within four hours, involve casualties whose conditions could worsen without swift evacuation. ‘Routine’ missions, lasting up to 24 h, involve stable casualties whose conditions are unlikely to deteriorate, thus permitting more flexibility in the scheduling of satellites and accommodating longer travel times for UXVs.

#### 3.2.2. The Data Latency Metric

The data latency metric assesses the time required for data transmission between the MEDCOM and the UXV, directly or via communication satellites, during the second phase of the MEDEVAC mission (see Figure 1b). The missions are classified into three latency classes: ‘Low’, ‘Real-time’, and ‘Non-real-time’. These classes dictate how the UXV can be controlled and the type of medical support it can provide (see Table 4). Neumeier et al. [46] indicate that a round-trip delay of 300 ms significantly impairs the driving performance of an autonomous vehicle traveling at a speed of 50 km/h, highlighting the necessity of low latency for the remote control of vehicles. Similarly, Nakamura et al. [47] determine that a one-way delay of 100 ms is still tolerable for such purposes. The International Telecommunication Union recommends a maximum one-way signaling delay of 100 ms for real-time tasks [48]. Drawing on these insights, the threshold for ‘Real-time’ latency is established at 100 ms for one-way transmission, defining the boundary between ‘Real-time’ and ‘Non-real-time’ classes.

Stricter latency limits are essential to precision medical operations, such as telesurgery. Nankaku et al. [49] find that delays of up to 100 ms are acceptable. However, they note that delays as short as 20 ms can still affect medical operations compared with no delay. Nonetheless, the difference between 20 ms and 50 ms is negligible, as both are within a range where skilled medical operators adapt effectively [50]. At 100 ms, delays significantly increase surgical errors and completion times. Experience is a substantial factor; experienced surgeons perform better at 100 ms latency than inexperienced surgeons at 0 ms latency. Building on these findings, the ‘Low’ latency class is set to one-way delays of 50 ms or less. Such low latency maintains a negligible impact of transmission delay on medical operations that require precision and responsiveness.

## 4. Simulation Results

The MEDEVAC mission is individually simulated for the FSS and each SSS. Each simulation is run under identical conditions, with the simulation time defined as 1 November 2024, from 08:00 to 20:00. This time frame is selected to represent a realistic operational window for executing the mission. The systems’ performance is compared by using the mission duration and data latency metrics. Additionally, the analysis includes evaluating the coverage, referring to the mission time frames when the imaging satellite makes visual contact with the casualty. Furthermore, the access of a satellite system, i.e., the mission time frames when the imaging satellite can communicate with the MEDCOM through ground stations, is considered.

### 4.1. Duration of MEDEVAC Mission

Figure 2 and Figure 3 provide the coverage and access schedules of the imaging satellites for the SSSs and the FSS, respectively. These schedules align perfectly, eliminating the need for ISLs, as data are transmitted immediately during the overlapping intervals. For each satellite system, a sum of the coverage and access schedules is also presented in Figure 2 and Figure 3. This sum combines the coverage and access of each satellite belonging to the specific satellite system, representing its overall capability. Some systems, like ‘France’ and ‘EU’, maintain relatively constant access to ground stations, while ‘Germany’ and ‘Italy’ exhibit more erratic behavior. The FSS has the most continuous access due to the large number of satellites and the enhanced connectivity from all satellites communicating with all ground stations. All SSSs have occasional coverage of the casualty at some point during the simulation period. Because of the combined effort of all satellites, the FSS naturally provides the best coverage.

The duration of the MEDEVAC mission, including the acquisition time for satellite images and the travel time of the UXV, is illustrated in Figure 4. The figure depicts how long the mission takes for each satellite system as a function of the mission’s start time. For example, if an imaging satellite approaches the casualty location shortly after the mission starts, the mission duration is brief. Conversely, if the mission begins after the satellite has passed the casualty, the mission duration can be considerably longer due to the time required for the next satellite to establish visual contact with the casualty. This situation is indicated by a spike in Figure 4. The durations of the MEDEVAC missions are compared across the classifications ‘Urgent’, ‘Priority’, and ‘Routine’ (see Table 3). The FSS consistently achieves the shortest mission duration for any start time by utilizing all available satellites.

The percentages of missions completed by each satellite system across the ‘Urgent’, ‘Priority’, and ‘Routine’ classes during the simulation are shown in Table 5. These data illustrate how frequently each system meets the required mission duration, emphasizing the performance differences between the FSS and the individual SSSs. If a satellite system can operate ‘Urgent’ missions, it inherently supports the less demanding ‘Priority’ and ‘Routine’ missions. Similarly, systems capable of ‘Priority’ missions can effectively manage ‘Routine’ missions. The ‘Routine’ mission capability is the most common for ‘Germany’, ‘Italy’, and ‘EU’. Notably, ‘France’ has the highest capability for ‘Priority’ missions for 39.7% of the simulation time and handles ‘Urgent’ missions for 22.1% of the time. ‘Germany’, ‘Italy’, and ‘EU’ cannot be relied upon for the ‘Urgent’ mission type, as they can conduct such missions for less than 10% of the simulation time. The ‘Federated’ system, which integrates all satellite systems, outperforms the others. It completes missions quickly due to uninterrupted coverage and superior access (see Figure 3). Its capabilities include 39.0% for ‘Urgent’, 49.4% for ‘Priority’, and 11.5% for ‘Routine’ missions.

### 4.2. Latency of Data Transmission

The latency of data transmission between the MEDCOM and the UXV is a crucial factor in the second phase of the MEDEVAC mission, as it directly affects the level of medical support that can be provided. The latency values, presented in Figure 5, highlight the significant differences among the satellite systems.

The ‘Federated’ and ‘EU’ systems, which utilize the OneWeb’s LEO satellite constellation, achieve a ‘Low’ classification, with a latency of approximately 14 ms. This low latency remains within the thresholds of 50 ms for telesurgery and 100 ms for the real-time navigation of the UXV. The lower orbital altitude and dense network of LEO satellites affirm that the ‘Federated’ and ‘EU’ systems can effectively manage critical MEDEVAC missions, where every second is vital to saving lives and preventing complications. In contrast, ‘France’, ‘Germany’, and ‘Italy’ rely on satellites like Syracuse, COMSAT, and Athena-Fidus, which exhibit ‘Non-real-time’ latency values ranging from 261 ms to 294 ms. These latency values exceed the acceptable limits for real-time medical operations, including telesurgery, rendering GEO-based systems less suitable for missions requiring precise control. The higher latency values stem from the longer transmission distances associated with the satellites’ higher orbital altitudes. Additionally, GEO satellites operate on single-satellite architectures, which limit redundancy and extend communication delays. As a result, these standalone satellite systems are better equipped for missions where low latency is not a priority, supporting non-time-sensitive tasks such as waypoint navigation for UXVs or the delivery of medical supplies.

## 5. Discussion

### 5.1. Interpretation of Simulation Results

The simulation results demonstrate that the FSS outperforms the SSSs in mission duration and data transmission latency. As summarized in Table 6, the ‘Federated’ system is the only one capable of supporting ‘Priority’ MEDEVAC missions while maintaining low data latency. The ‘EU’ system also achieves low latency but is effective only for ‘Routine’ missions. While the ‘France’ system can support ‘Priority’ missions, it cannot maintain low latency. The ‘Federated’ system consistently sustains its capability for ‘Urgent’ and ‘Priority’ missions with only minor gaps. An example of such a lapse is shown in Figure 4 at 11:30, where the system briefly fails to meet the requirements of the ‘Priority’ mission fully. Multiple satellite systems ensure reliable and rapid updates, significantly reducing mission duration. LEO-based systems, including ‘Federated’ and ‘EU’, utilize the OneWeb constellation to achieve low latency, making them well suited for time-sensitive missions that require real-time control and quick medical support. In contrast, unlike the’ France’ system, most GEO-based systems, namely, ‘Germany’ and ‘Italy’, are suitable for ‘Routine’ missions due to their higher transmission delays.

The simulations reveal that ISLs are unnecessary for the mission presented in this study, as the imaging satellite’s coverage and access schedules are already well aligned. This is a significant finding, given that ISLs represent a core feature of the FSS. However, ISLs could become essential if the ground stations are farther apart or the mission occurs in a more remote region.

### 5.2. Implications of the Simulation Results

Integrating UXVs with satellite systems for MEDEVAC missions offers transformative potential on the modern battlefield. An FSS enables low-latency communication and shorter mission durations, which are crucial to time-sensitive missions like casualty evacuation. The near-continuous geospatial coverage (see Figure 3) enhances the efficiency of medical evacuations, improving the survival rates of injured personnel. In particular, the LEO satellite constellations support real-time medical interventions such as telesurgery and enable precise navigation of UXVs in high-risk missions.

Despite the advantages of an FSS, implementing such a system for a MEDEVAC mission presents several challenges. Key obstacles include the integration of ISLs, the establishment of unified communication protocols between satellite systems, and the implementation of robust data security measures. These challenges require extensive collaboration across organizations and nations. NATO’s APSS is a notable example, aiming to integrate governmental and commercial satellite assets into a cohesive network. The anticipated cost of EUR 1 billion for APSS and its technical requirements underscore significant obstacles to adopting FSSs [20]. However, these challenges are expected to lessen by 2030 due to advancements in satellite networks and inter-satellite communication [51]. Nevertheless, such systems have the potential to become essential to multinational MEDEVAC missions, particularly in contested environments where secure communication and consistent satellite availability are critical.

The findings of this paper show that current SSSs struggle to support MEDEVAC operations (see Table 6). To effectively support MEDEVAC missions, the satellite systems must deliver continuous coverage, rapid revisit times, and reliable real-time data transfer. Additionally, they must address the specific requirements of the missions, including low-latency communication for the control of UXVs, high-resolution imaging for navigation, and secure data transmission to protect mission integrity. Similarly, UXVs must evolve to meet the demands of future MEDEVAC missions. This includes integrating robust communication systems to maintain uninterrupted connectivity with satellites and the MEDCOM, particularly for missions utilizing telemedicine capabilities.

### 5.3. Limitations

While the simulation experiment provides valuable insights into integrating satellites and UXVs for MEDEVAC missions, it is subject to a few limitations and may not fully capture real-world complexities. The simulation model overlooks possible electronic warfare threats, such as jamming or network failures, which could hinder satellite–UXV communication. Additionally, it does not consider the impact of GNSS-denied environments, where GPS signals may be blocked, spoofed, or degraded. The deployment of an FSS also presents potential security risks, as collaboration across organizations and countries might heighten the likelihood of adversaries intercepting or manipulating data transmission. Moreover, political and military tensions among nations could further impact the availability and reliability of shared satellite resources, potentially restricting access to crucial FSS capabilities in conflict situations. Measurements conducted in Finland indicate that the latency of LEO satellite systems is comparable to that of 5G mobile networks when both the mobile phone and the satellite terminal are in motion. With current systems, the satellite antenna needs to be larger than the 5G one. A significant advantage of using a satellite system is the connectivity in remote areas, where terrestrial base stations do not always exist. A further limitation is that the study simulates only one scenario, which may not fully reflect the response capabilities of SSSs and FSSs under different operational conditions. Finally, the MEDEVAC missions examined in the simulation experiment do not extend beyond the time the UXV reaches the casualty. The findings of this paper should be understood considering these limitations and assumptions, emphasizing the need for further research, including enhancements to the simulation model.

### 5.4. Future Directions

As the integration of satellite systems and UXVs for MEDEVAC missions continues to evolve, future research should address several areas to improve the simulation model, conduct field tests, and enhance satellite and UXV technology. The extended version of the simulation model should consider data transfer issues such as delays, jitter, and packet loss to capture the complexities of real-world MEDEVAC missions better. Additionally, incorporating satellite outages and ISL disruptions would help simulate realistic telesurgery and UXV control conditions. Extending the simulation period beyond 12 h would allow for a more detailed assessment of long-term satellite availability, enabling a more accurate analysis of prolonged missions. Furthermore, scaling the model to include multiple simultaneous missions across different regions will provide a more comprehensive view of the satellite system’s ability to support large-scale missions.

In addition to simulation, conducting real-world field testing would be beneficial for validating the integration of satellites and UXVs. While simulations help assess scalability by including various mission types, longer evacuation distances, and multiple operation regions, they cannot fully replicate real-world challenges like weather, terrain, and communication issues. Large-scale field testing, namely, military exercises, would provide empirical data on how the satellite–UXV system performs under actual conditions, including coordination with helicopters, medical teams, and ground forces. Testing in stressful situations, like satellite outages or severe weather, is essential to refining the system’s design and ensuring reliable performance in unpredictable combat environments.

Future research should also examine the standardization of tasking protocols. This encompasses defining command structures, data exchange methods, task assignment processes, error handling, mission timelines, and sequencing between satellites and UXVs in MEDEVAC missions. Standardizing tasking protocols facilitates smooth communication and coordination. As autonomous vehicles, drones, and robotic systems become increasingly common, interoperability among diverse systems will be critical to mission success. Standardized protocols would allow satellites to prioritize and manage UXV tasks in real time, ensuring seamless integration across platforms and minimizing the risk of delays or failures. These protocols are essential to large-scale missions that involve multiple systems, enhancing coordination and reducing the duration of MEDEVAC missions. Additionally, future research should incorporate the effects of electronic interference and real-world battlefield conditions on satellite and UXV communications to address operational challenges better and improve these systems’ resilience in contested environments.

## 6. Conclusions

This paper examined the role of an FSS in enhancing UXV-supported MEDEVAC missions through a simulation-based experiment comparing one FSS-enabled mission with four using SSSs. To the authors’ knowledge, this was the first experiment to explore the integration of the FSS in such military missions. A simulation model was developed by using STK software to assess the satellite systems based on mission duration, data latency, coverage, and satellite access. The simulation results demonstrated the operational advantages of the FSS in the MEDEVAC missions, particularly its ability to provide more reliable satellite imaging coverage and reduce mission duration. The low latency of LEO satellites within the FSS enabled real-time navigation of UXVs and telesurgery—essential capabilities for time-sensitive battlefield evacuations. In contrast, the GEO-based systems commonly used in SSSs exhibited higher latency, which limits their effectiveness in high-risk environments. Overall, the findings of this paper suggest that investing in FSSs and fostering international collaboration are crucial efforts for ensuring scalability, interoperability, and frequent-imaging opportunities—key features for effectively using satellite systems to support UXV-based MEDEVAC missions.

While the simulations conducted in this paper highlighted the superiority of FSSs in supporting UXVs during MEDEVAC missions, they also revealed challenges that must be addressed. Future research should focus on standardized tasking protocols, robust ISLs, and secure communication frameworks to ensure seamless coordination between satellites and UXVs. On the other hand, the developed simulation model offers a valuable foundation for further research on integrating FSSs and UXVs in MEDEVAC missions. Promising extensions of this model include assessing satellite failures, electronic warfare threats, and atmospheric interference. Additionally, field testing is essential to validating satellite–UXV collaboration not only in military evacuation tasks but also, e.g., in humanitarian activities.

## Figures and Tables

**Figure 1 sensors-25-01655-f001:**
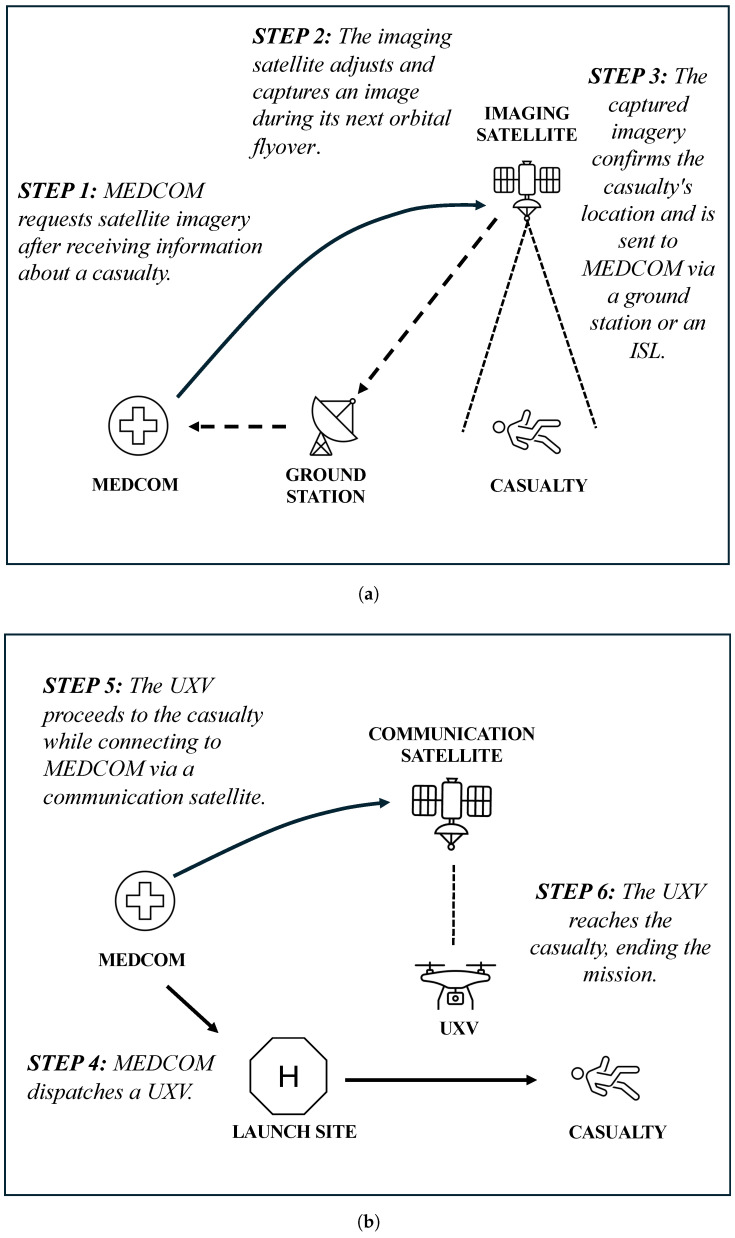
The MEDEVAC mission. (**a**) The first phase begins with the MEDCOM requesting satellite imagery to confirm the casualty’s location. The imagery is captured and then transmitted via a ground station or an ISL. (**b**) The second phase involves the MEDCOM dispatching a UXV while monitoring it in real time. When the UXV reaches the casualty, the MEDEVAC mission ends in the simulation experiment.

**Figure 2 sensors-25-01655-f002:**
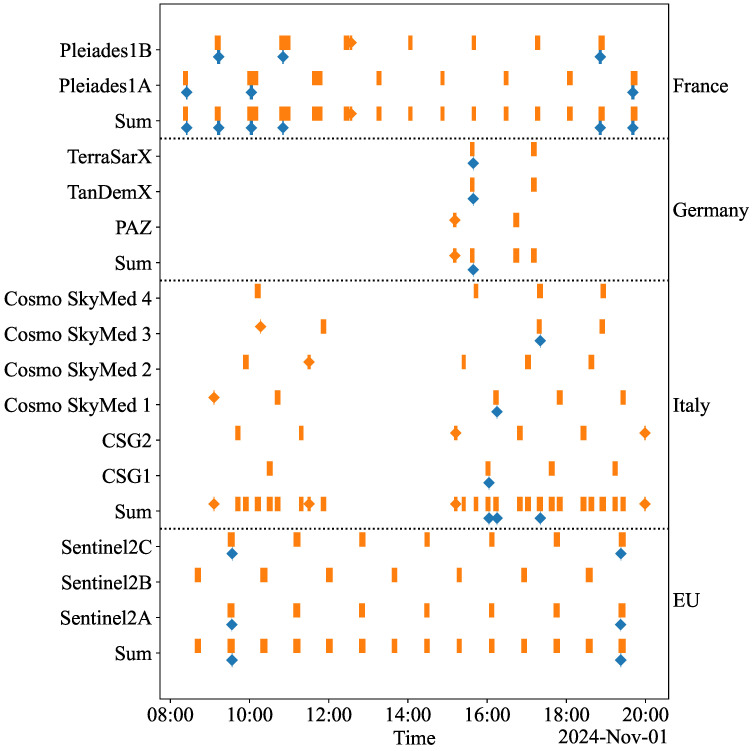
The coverage and access schedules for each SSS. Blue intervals indicate coverage periods, while orange intervals represent access periods. Sum refers to the aggregated coverage and access periods of the satellites.

**Figure 3 sensors-25-01655-f003:**
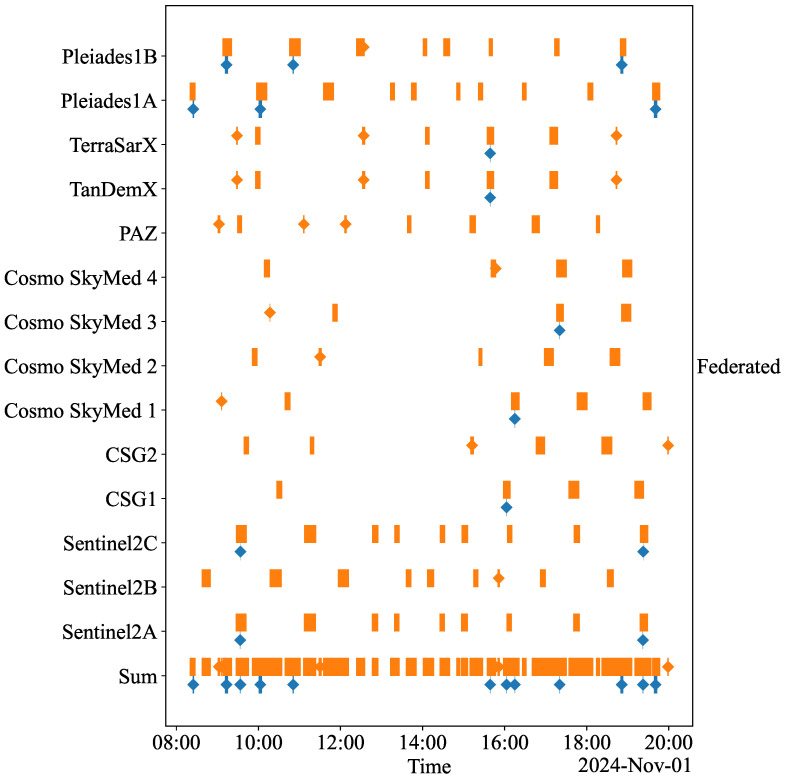
The coverage and access schedules for the FSS. Blue intervals indicate coverage periods, while orange intervals represent access periods. Sum refers to the aggregated coverage and access periods of the satellites.

**Figure 4 sensors-25-01655-f004:**
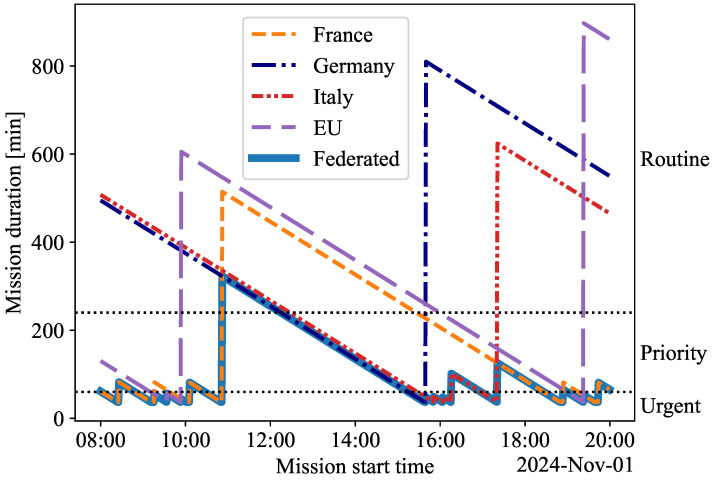
The mission duration for ‘France’, ‘Germany’, ‘Italy’, ‘EU’, and ‘Federated’ systems as a function of the start time of the mission. Dotted horizontal lines represent the limits between the classes of mission duration.

**Figure 5 sensors-25-01655-f005:**
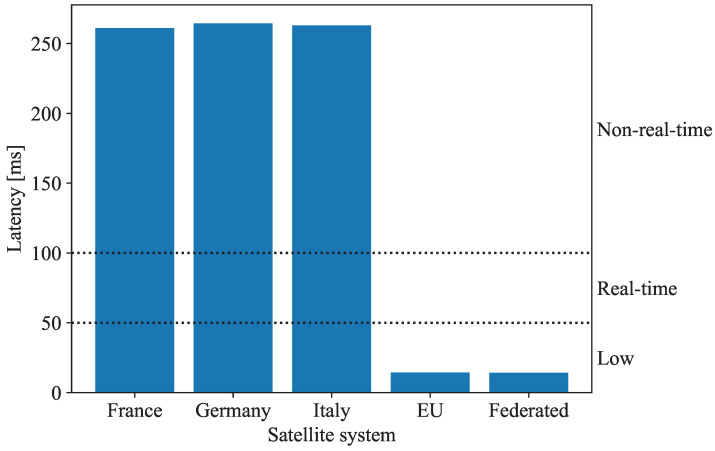
Latency classifications for the ‘France’, ‘Germany’, ‘Italy’, ‘EU’, and ‘Federated’ systems.

**Table 1 sensors-25-01655-t001:** The key differences between an FSS and an SSS.

Feature	Federated Satellite System	Standalone Satellite System
Architecture	A network of interconnected satellites linked through ISLs.	Independent satellites with no interlinking.
Data sharing	Authenticated users gain access to data from all satellites within the federation.	Authenticated users access data from a single satellite system.
Coverage	Provides a wider coverage area by leveraging multiple satellite constellations.	Limited to the coverage area of a standalone satellite system.
Revisit times	Reduced revisit times due to multiple satellites.	Longer revisit times dependent on the availability of individual satellites.
Communication speed	Faster, as data can be directly relayed to the target location through ISLs.	Slower, relying on a direct ground station link.
Payload resource	Payload is shared across the satellite network.	Restricted to the independent capability of each satellite.
Operational flexibility	High flexibility, allowing for data rerouting and imaging tasks across the entire satellite network.	Limited flexibility, with satellites operating independently.
Resilience to coverage gaps	Highly resilient, with additional constellations compensating for coverage gaps.	Vulnerable to gaps due to a limited number of satellites.
System complexity	Higher complexity due to inter-satellite coordination among multiple organizations.	Simpler, with less need for cross-organizational management.

**Table 2 sensors-25-01655-t002:** The description of the satellite systems used in the simulation experiment.

Satellite System	Imaging Satellites		Communication Satellites
	Satellites	Ground Stations	
France	Pleiades1A Pleiades1B	Toulouse, France Kiruna, Sweden	Syracuse3A Syracuse3B
Germany	TanDemX TerraSarX PAZ	Weilheim, Germany Neustrelitz, Germany	ComSatBw1 ComSatBw2
Italy	CosmoSkymed1 CosmoSkymed2 CosmoSkymed3 CosmoSkymed4 CosmoSkymed4 CSG2	Cordoba, Argentina Kiruna, Sweden	Athena-Fidus
EU	Sentinel2A Sentinel2B Sentinel2C	Kiruna, Sweden	OneWeb constellation
Federated	France Germany Italy EU	Uses the closest ground station available	Uses the fastest communication satellite available

**Table 3 sensors-25-01655-t003:** Classification of MEDEVAC missions based on mission duration.

Classification	Duration of Mission (h)	Description
Urgent	0–1	Casualties needing urgent evacuation within one hour must be quickly treated to prevent serious complications or permanent damage.
Priority	1–4	Casualties needing evacuation within four hours require prompt action to prevent worsening conditions or increased suffering.
Routine	4–24	Casualties needing evacuation within 24 h have stable conditions and are unlikely to worsen.

**Table 4 sensors-25-01655-t004:** Classification of MEDEVAC missions based on data latency metric.

Classification	Latency (ms)	Description
Low	0–50	Low latency enables telesurgery and all activities included in the ‘Real-time’ and ‘Non-real-time’ classes.
Real-time	50–100	Real-time latency enables the steering of the UXV and all activities included in the ‘Non-real-time’ class.
Non-real-time	>100	Non-real-time latency enables the evacuation of a casualty such that the route of the UXV is determined prior to the dispatch of the UXV.

**Table 5 sensors-25-01655-t005:** The percentages of missions completed by each satellite system across the ‘Urgent’, ‘Priority’, and ‘Routine’ classes.

Satellite System	Capability for Urgent Missions (%)	Capability for Priority Missions (%)	Capability for Routine Missions (%)
France	22.1	39.7	38.2
Germany	3.5	25.0	71.5
Italy	9.9	30.8	59.3
EU	9.2	34.9	56.0
Federated	39.0	49.4	11.5

**Table 6 sensors-25-01655-t006:** Summary of simulation results.

Satellite System	The Most Suitable MEDEVAC Mission Type	Capability for Low Latency
France	Priority	No
Germany	Routine	No
Italy	Routine	No
EU	Routine	Yes
Federated	Priority	Yes

## Data Availability

The original contributions presented in this study are included in the article. Further inquiries can be directed to the corresponding author.

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
