# Peer review of "Leveraging Federated Satellite Systems for Unmanned Medical Evacuation on the Battlefield"

_sensors, 2025, doi:10.3390/s25061655_

Round 1
Reviewer 1 Report
Comments and Suggestions for Authors
I would suggest the authors improve the resolution and readability of the figures such as figure 1
Are there any security risks to deploying the FSS method? A brief mention of this would be a good addition to the Introduction section or the “limitation” section
The authors mention possible limitations such as spoofing, GNSS denied aeras and network failures. How about political and military actions between nations? Can this affect the feasibility of the FSS? If yes, mention this a possible limitation
The authors have used over 10 references that are 2015 or before that. Are there any updated similar studies that can be used in this reference list?
A bibliometric analysis of the spread of research on this topic in the past decade would be a good addition to the introduction/review section.
A figure on the workings of FSS would also be useful to the reader to better visualize the process. This figure can be placed in section1.
Author Response
Response to the Reviewer’s comments on the manuscript entitled ‘Leveraging federated satellite systems for unmanned medical evacuation on the battlefield,’ Manuscript Sensors-3504396
We want to thank the Reviewer for her/his comments. We have carefully studied the comments, and our responses are presented below. A marked copy of our manuscript has been uploaded, and the added/changed text has been highlighted in red font. A clean copy of the manuscript has also been uploaded.
- Comment 1: I would suggest the authors improve the resolution and readability of the figures such as figure 1.
- Response 1: We appreciate the reviewer's suggestion regarding the figures. To improve readability, we have enhanced the clarity of Figures 1 and 4. Alternative presentation methods for the figures were considered, but it is difficult without losing too much data.
- Comment 2: Are there any security risks to deploying the FSS method? A brief mention of this would be a good addition to the Introduction section or the “limitation” section.
- Response 2: The reviewer raises an important point. To provide a more comprehensive overview, we have enhanced the discussion on potential security risks associated with deploying the FSS method in the Limitations section.
- Comment 3: The authors mention possible limitations such as spoofing, GNSS denied aeras and network failures. How about political and military actions between nations? Can this affect the feasibility of the FSS? If yes, mention this a possible limitation.
- Response 3: Geopolitical factors, including international conflicts, regulatory restrictions, and strategic military interests, can indeed influence the feasibility of FSS operations. The Limitations section considers these factors now.
- Comment 4: The authors have used over 10 references that are 2015 or before that. Are there any updated similar studies that can be used in this reference list?
- Response 4: We acknowledge the concern regarding older references. However, these are relevant original papers that form the foundation of our study and are therefore essential references. Where applicable, we have also incorporated more recent studies to ensure a balanced and up-to-date perspective.
- Comment 5: A bibliometric analysis of the spread of research on this topic in the past decade would be a good addition to the introduction/review section.
- Response 5: While a bibliometric analysis could offer insights into research trends, our study concentrates on the practical and operational aspects of FSS-enhanced MEDEVAC rather than a broad examination of publication trends regarding satellite systems. Our manuscript is already extensively linked to the essential existing literature related to the topic, ensuring that our analysis is well-grounded in relevant research. In our opinion, a more in-depth literature review, including a bibliometric analysis, would only unnecessarily increase the length of the manuscript without adding relevant references to the text. Instead, we have conducted a targeted review of the most appropriate and recent literature to maintain a focused discussion aligned with our research objectives.
- Comment 6: A figure on the workings of FSS would also be useful to the reader to better visualize the process. This figure can be placed in section1.
- Response 6: We appreciate the suggestion. However, Section 1 already provides a detailed textual explanation of the FSS principles. Additionally, Table 1 illustrates the key aspects of FSS and their differences to SSS. We believe an additional figure is unnecessary to maintain conciseness and avoid redundancy.
Reviewer 2 Report
Comments and Suggestions for Authors
-
Terms appearing for the first time in the abstract and the main text should be provided with both their full names and abbreviations, such as "medical evacuation" in the abstract. Please check if similar issues exist in the article.
-
The paper mentions that FSS requires collaboration across organizations and countries, which may lead to the risk of data transmission being intercepted by adversaries. Please explain how data security will be ensured.
-
The article only simulates one scenario, which does not fully reflect the response capabilities of SSS and FSS in different scenarios.
-
Some figures (such as Figure 4) lack clarity in certain details, especially where lines overlap and are thin, making them difficult to distinguish. It is suggested to thicken the lines to make the data in the charts easier to observe and analyze.
-
The article mentions that the data delay for FSS is 50ms, while for SSS it is around 200ms, which may affect real-time medical applications. It is recommended to compare the advantages of this method with technologies like 5G communication to determine if it performs better.
-
The simulation experiments in the paper do not consider electronic interference and the impact of real-world conditions on satellite and drone communications. It is suggested to include these factors in future research to more accurately simulate the communication challenges in battlefield environments.
Author Response
Response to the Reviewer’s comments on the manuscript entitled ‘Leveraging federated satellite systems for unmanned medical evacuation on the battlefield,’ Manuscript Sensors-3504396
We want to thank the Reviewer for her/his comments. We have carefully studied the comments, and our responses are presented below. A marked copy of our manuscript has been uploaded, and the added/changed text has been highlighted in red font. A clean copy of the manuscript has also been uploaded.
- Comment 1: Terms appearing for the first time in the abstract and the main text should be provided with both their full names and abbreviations, such as "medical evacuation" in the abstract. Please check if similar issues exist in the article.
- Response 1: We appreciate the reviewer's suggestion. We have now made the necessary changes to the abstract to ensure that terms appearing for the first time, such as "medical evacuation," are provided with their full names and abbreviations. However, we have carefully reviewed the main text and confirmed that similar issues do not exist.
- Comment 2: The paper mentions that FSS requires collaboration across organizations and countries, which may lead to the risk of data transmission being intercepted by adversaries. Please explain how data security will be ensured.
- Response 2: The reviewer raises an important point. To provide a more comprehensive overview, we have enhanced the discussion on potential security risks associated with deploying the FSS in the Limitations section.
- Comment 3: The article only simulates one scenario, which does not fully reflect the response capabilities of SSS and FSS in different scenarios.
- Response 3: We acknowledge that simulating only one scenario may not capture the full range of response capabilities of SSS and FSS in different operational conditions. However, our study was designed to provide an initial, focused assessment of their impact in a well-defined use case. The results and conclusions drawn from this scenario remain valid within its intended scope. One of the reasons we did not include additional scenarios was to maintain a clear and concise manuscript, as adding scenarios would have unnecessarily lengthened the text without significantly altering the key findings. Nevertheless, we recognize this as a limitation and have highlighted the need for future studies to explore multiple scenarios – please see sections Limitations and Future Directions.
- Comment 4: Some figures (such as Figure 4) lack clarity in certain details, especially where lines overlap and are thin, making them difficult to distinguish. It is suggested to thicken the lines be thickened to make the data in the charts easier to observe and analyze.
- Response 4: We appreciate the reviewer's suggestion regarding the figures. To improve readability, we have enhanced the clarity of Figures 1 and 4. Alternative presentation methods for the figures were considered, but it is difficult without losing too much data.
- Comment 5: The article mentions that the data delay for FSS is 50ms, while for SSS it is around 200ms, which may affect real-time medical applications. It is recommended to compare the advantages of this method with technologies like 5G communication to determine if it performs better.
- Response 5: We agree that a comparison to a 5G system is generally valuable in understanding the performance of satellite systems. We have added the following sentences in the text: “Measurements conducted in Finland indicate that the latency of LEO satellite systems is comparable to that of 5G mobile networks when both the mobile phone and the satellite terminal are in motion. With current systems, the satellite antenna needs to be larger than the 5G one. A significant advantage of using a satellite system is the connectivity in remote areas where terrestrial base stations do not always exist."
- Comment 6: The simulation experiments in the paper do not consider electronic interference and the impact of real-world conditions on satellite and drone communications. It is suggested to include these factors in future research to simulate the communication challenges in battlefield environments more accuratel.
- Response 6: We agree that the impact of electronic interference and real-world conditions on satellite and drone communications is an essential factor. While our current study does not incorporate these aspects, we acknowledge this limitation and suggest incorporating electronic warfare considerations in future research to better reflect battlefield communication challenges. This is now mentioned in the Future Directions section.
Round 2
Reviewer 1 Report
Comments and Suggestions for Authors
All my comments have been addressed.